# Flexible three-dimensional artificial synapse networks with correlated learning and trainable memory capability

Chaoxing Wu [1], Tae Whan Kim[1], Hwan Young Choi[1], Dmitri B. Strukov[2] & J. Joshua Yang [3]

If a three-dimensional physical electronic system emulating synapse networks could be built, that would be a significant step toward neuromorphic computing. However, the fabrication complexity of complementary metal-oxide-semiconductor architectures impedes the achievement of three-dimensional interconnectivity, high-device density, or flexibility. Here we report flexible three-dimensional artificial chemical synapse networks, in which two-terminal memristive devices, namely, electronic synapses (e-synapses), are connected by vertically stacking crossbar electrodes. The e-synapses resemble the key features of biological synapses: unilateral connection, long-term potentiation/depression, a spike-timing-dependent plasticity learning rule, paired-pulse facilitation, and ultralow-power consumption. The three-dimensional artificial synapse networks enable a direct emulation of correlated learning and trainable memory capability with strong tolerances to input faults and variations, which shows the feasibility of using them in futuristic electronic devices and can provide a physical platform for the realization of smart memories and machine learning and for operation of the complex algorithms involving hierarchical neural networks.

[1] Department of Electronic and Computer Engineering, Hanyang University, Seoul 133-791, Korea. [2] Department of Electrical and Computer Engineering, University of California at Santa Barbara, Santa Barbara, CA 93106, USA. [3] Department of Electrical and Computer Engineering, University of Massachusetts, Amherst, MA 01003-9292, USA. Chaoxing Wu and Tae Whan Kim contributed equally to this work. Correspondence and requests for materials should be addressed to T.W.K. (email: twk@hanyang.ac.kr)

The brain is able to remember, learn, and process multi-dimensional information through an energy-efficient and fault-tolerant computation process. As a result, the idea of building an electronic system that can mimic the function of the brain is currently attracting significant interest[1, 2]. Note that as many as $10^{14}$ synapses are present in the human cerebral cortex[3], making the hardware implementation with three-dimensional (3D), massively-parallel, and compact electronic systems exceptionally challenging due to the lack of a compact electronic element. Recently, two-terminal and three-terminal electronic devices with tunable resistance have been widely demonstrated in the pursuit of certain synaptic functions, with the device's conductance representing the synaptic weight[4–9]. Especially, synaptic operations, including long-term potentiation/depression (LTP/LTD), short-term potentiation/depression, a spike-timing-dependent plasticity (STDP) learning rule, paired-pulse facilitation (PPF), and low-power consumption, have been extensively simulated or emulated in a single device[10–21]. The recent implementation of a flat-panel array with a partial neuromorphic function is an example of the exciting progress that is being made[22–30]. However, the 3D interconnectivity, which plays a vital role in high-density information storage and multi-dimensional information processing in biological neural networks, has not yet been well realized with existing electronic devices[31]. This constitutes an obstacle to the practical applications of artificial-neural-network devices.

In the semiconductor electronics field, the multilayer stacking architecture with a crossbar structure is an excellent candidate for realizing 3D interconnectivity. However, unintentional current leakage paths can result in a misreading within the cells, namely, the crosstalk effect[32, 33]. As a result, the electronic component in the crossbar structure should be connected to a selector device to suppress that effect, but that would lead to a decrease in the integration level of the array and to an increase in the complexity of 3D interconnectivity. Even though synaptic plasticity has already been extensively demonstrated, the use of one inherent characteristic of a chemical synapse, the one-direction transmission of signals, has rarely been reported. At a chemical synapse, one pre-synaptic cell releases neurotransmitter molecules into the synaptic cleft that is adjacent to another cell; then, these molecules bind to receptors on the side of the post-synaptic cell of the synaptic cleft[34]. This means that the chemical synapses pass information directionally from a pre-synaptic cell to a post-synaptic cell, which is similar to the rectification behavior of a rectifier diode and provides a potential solution to suppress the crosstalk effect.

On the other hand, the utilization of inorganic functional layers is also an impediment to the realization of high-performance flexibility. For typical organic materials, the lower functional layers might be partially re-dissolved during the deposition process or the lithography processes of the upper layers, which makes the preparation of a multilayered stacking structure very challenging. It is worth noting that poly(methylsilsesquioxane) (pMSSQ) has excellent flexibility, as well as good thermal, chemical, and physical stabilities, for use in flexible electronics (Supplementary Fig. 1)[35, 36]. The fully cross-linked structure of pMSSQ allows the simple 3D stacking structure to be fabricated layer by layer through a solution processable approach without the problem of re-dissolving the previously deposited lower layers.

Here we report a flexible, 3D stacking, artificial chemical synapse network (3D-ASN) by utilizing selector-device-free electronic synapses (e-synapses). The e-synapses based on Cu-doped pMSSQ resemble the key features of biological synapses, with LTP/LTD, a STDP learning rule, PPF learning, and ultralow-power consumption (in the pJ range for one spike). On the basis of the rectification characteristic of e-synapse, the crosstalk effect can be suppressed, which makes the selector-device unnecessary and simplifies the device structure. Moreover, the 3D-ASN is shown to be able to mimic correlated learning and exhibit a trainable memory function with a strong tolerance to input faults; this is accomplished by taking advantage of the programmable synaptic weights of the e-synapses.

## Results

**Device structure and electrical behaviors.** A conceptual schematic diagram and a photograph of the flexible 3D-ASN based on copper-ion-doped pMSSQ (Cu-pMSSQ) are presented in Fig. 1a and Fig. 1b, respectively. Top electrodes and bottom electrodes cross each other perpendicularly in each layer of the crossbar structure, and the active neuromorphic memory layer is sandwiched between the electrodes, as shown in the cross-sectional scanning electron microscope (SEM) image in Fig. 1c. The Cu-pMSSQ composites are used as the neuromorphic medium. A schematic diagram of an e-synapse formed at each intersection in the crossbar with a sandwiched structure of Al (bottom electrode)/lightly doped layer/highly doped layer/Al (top electrode) is shown in Fig. 1d.

The nature of the resistive switching in the memristive device is still an important subject under active studies. Different models have been suggested, including the alteration of the bulk insulator's resistivity due to the migration of ions or to a trapping-releasing process of carriers[37, 38], the modification of the metal/insulator interface's resistivity[39], and the formation of localized metal-atom chains bridging the electrode materials[26, 40]. Here, we attribute the conductance changes of our e-synapse to the migration of ions induced by input impulses. The pMSSQ-based hybrid polymer can act as a hole-injection material, as was previously demonstrated[36]. We further find that the doping of Cu ions can modulate the electric conductance of pMSSQ, which is consistent with a previously reported result on Cu-doped polymer[41]. The insulating pMSSQ layer can be transformed into a semiconducting layer by being doped with a high concentration of copper ions (Supplementary Fig. 2a). Ultraviolet–visible (UV–Vis) absorption spectra of Cu-doped pMSSQ layers with different Cu concentrations show that the energy gap decreases with increasing Cu concentration (Supplementary Fig. 2b). UV photoemission spectroscopy (UPS) spectra show that doping Cu ions shifts the highest occupied molecular orbital (HOMO) of Cu-doped pMSSQ to a lower binding energy, which can reduce the barrier to hole injection from the electrode to the functional material (Supplementary Fig. 2c). As a result, when the top electrode is forward-biased, the current is relatively large because a large number of carriers can overcome the low barrier between the Al Fermi level and the HOMO of the Cu-doped layer through Fowler–Nordheim tunneling (Fig. 1d). However, the electric field will simultaneously drive copper ions away from the interface with the top electrode, which increases the barrier height (Supplementary Fig. 3). This larger barrier height will reduce the current, resulting in a decrease in the conduction of the e-synapse. When a negative bias is applied to the top electrode, thermionic emission dominates carrier transport in the device in the low-voltage region due to the higher barrier between the Al and the lightly doped Cu-doped layer (Fig. 1d and Supplementary Fig. 4). As a result, the reverse current is much smaller than the forward current, which leads to the rectification characteristic of our e-synapse. The electric field will also drive copper ions away from the interface with the bottom electrode and attract copper ions toward the interface with the top electrode at the same time, which decreases the barrier height at the interface with the top electrode (from the *dashed line* to the *solid line*). As a result, a sufficient negative bias drives the copper ions to the interface with the top electrode, allowing the copper concentration to reach its initial high level.

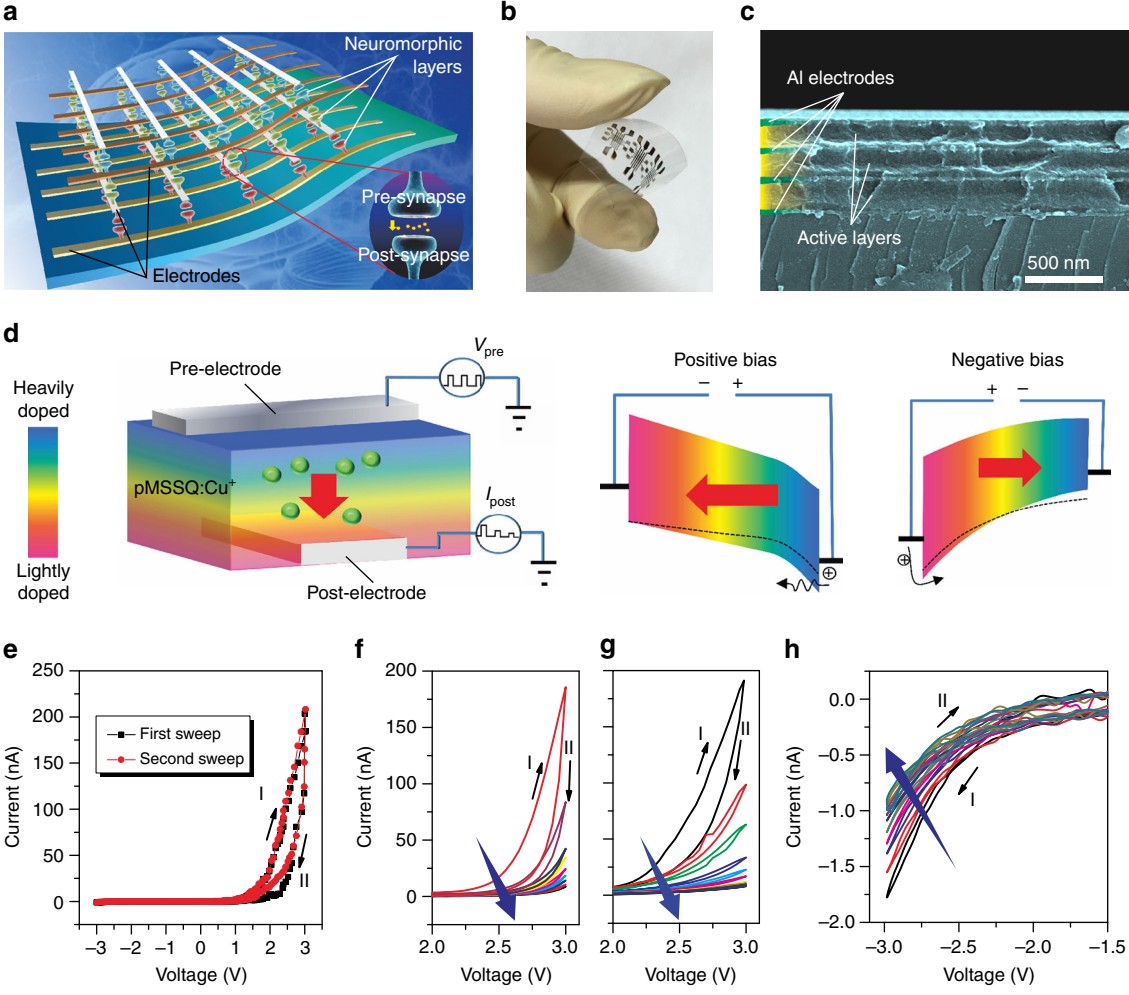

**Fig. 1** Device structure and electrical behaviors of 3D-ASN. Illustration (**a**), photograph (**b**), and cross-sectional SEM image (**c**) of the flexible 3D-ASN based on e-synapses. The active neuromorphic memory layer is sandwiched between the top electrode and the bottom electrode, thus forming an e-synapse. The top and the bottom electrodes correspond to the pre-synaptic and the post-synaptic neurons, respectively. **d** Schematic diagram and band structure of the e-synapse consisting of a sandwiched structure of Al (bottom electrode)/lightly doped layer/highly doped layer/Al (top electrode). Under positive bias, the Fowler–Nordheim tunneling process dominates the carrier conduction, and the electric field attracts copper ions away from the interface with the top electrode, resulting in an increase in the barrier height at that interface (from the *dashed line* to the *solid line*). Under negative bias, the thermal emission process dominates the carrier conduction of the e-synapse, and the electric field attracts copper ions toward the interface with the top electrode, resulting in a decrease in the barrier height at that interface (from the *dashed line* to the *solid line*). **e** *I–V* curves of the e-synapse under dual voltage sweeping from −3 to 3 V. The *I–V* curves shows a rectification characteristic with a resumable hysteresis. **f** *I–V* curves under consecutive positive voltage sweeps. **g** *I–V* curves under consecutive positive voltage sweeps after the negative voltage sweeps in **h**. **h** *I–V* curves under consecutive negative voltage sweeps after the positive voltage sweeps in **f**. The e-synapse can be repeatedly programmed from the low-resistance to the high-resistance states and from the high-resistance to the low-resistance states

As shown in Fig. 1e, *I–V* curve of a single e-synapse is asymmetric with a resumable hysteresis and a remarkable rectification characteristic. The resumable hysteresis results from the movement of the copper cations mentioned above. One should note that the rectification characteristic is rarely considered in reported artificial synapses. However, when the crossbar memory array is integrated, the significant advantage of the rectification characteristic of our e-synapses can be demonstrated, which will be discussed later. Figure 1f shows the direct-current characteristics of our e-synapse. The dual *I–V* sweep (0 → 3 → 0 V) shows clockwise hysteresis. While the e-synapse under a forward sweep is in a low-resistance state, under a reverse sweep, it switches into a relatively high-resistance state. When consecutive positive voltage sweeps are applied, the conductivity of the e-synapse continuously decreases. After the positive voltage sweeps, negative voltage sweeps (0 → −3 → 0 V) are applied, and the results are shown in

Fig. 1h. The conductivity of the e-synapse continuously decreases to a stable value with increasing number of voltage sweeps. Interestingly, after a sufficient number of negative biases are applied, the e-synapse can recover its original highly conducting state and its original electrical behaviors under positive bias, as shown in Fig. 1g. Consequently, the e-synapse can be repeatedly programmed, including the process of going from the low-resistance to the high-resistance state and the process of going from the high-resistance to the low-resistance state.

**Ultra-low-power LTP/LTD.** The realization of variations in the synaptic weight by using an e-synapse is believed to be the most important step toward realizing other complex neurological functions through the use of neuromorphic electronics. Figure 2a shows the results for an e-synapse programmed by using a series

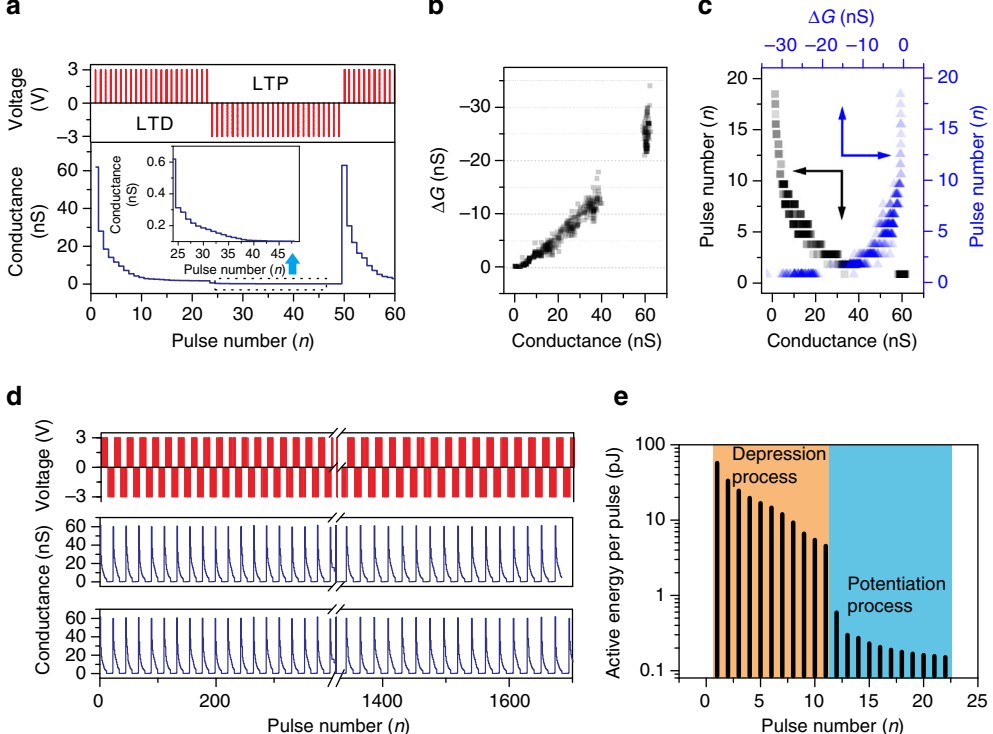

**Fig. 2** Ultra-low-power LTP/LTD. **a** Switching behaviors under a sequence of electric pulses. The *inset* presents enlarged view in the range from 25 to 50 pulses. LTD stimulation: a series of 3-V, 0.1-ms pulses are applied, which can decrease the conductance of the e-synapse. LTP stimulation: a series of −3-V, 0.1-ms pulses are applied, which can recover the high conductance of the e-synapse. **b** Conductance variation (Δ$G$) under a 3-V pulse (0.1 ms) as a function of the initial conductance ($G$) during depression, which encompasses data from 1000 measurements. **c** Number of positive pulses (3 V, 0.1 ms) applied to the e-synapse as a function of the initial conductance ($G$) and as a function of the measured Δ$G$, which encompasses data from 1000 measurements. For example, for $G = 20$ nS, the number of the applied positive pulses is between three and four, while for Δ$G = −5$ nS, the number of applied positive pulses is between four and six. **d** Potentiation–depression cycling stress tests when the e-synapse is in a flat state (*middle panel*) and in a curved state with a radius of surface curvature of 10 mm (*bottom panel*). **e** Energy consumption per pulse (±3 V, 0.1 ms) in both the depression and the potentiation processes

of 25 identical positive pulses (3 V, 0.1 ms), followed by a series of 25 identical negative voltage pulses (−3 V, 0.1 ms). The conductance ($G$), namely the synaptic weight, gradually decreases with increasing number of positive pulses. Interestingly, the change in the synaptic weight is nonvolatile (Supplementary Fig. 5), which means that the change in the synaptic weight under positive pulses exhibits LTD plasticity. It is worth noting that the conductance of the e-synapse should be measured under a positive bias because of its remarkable rectification characteristics and negligible conductance under a negative bias. After the application of sufficient negative pulses, the e-synapse returns to its original high-conductance state measured under positive pulses. This means that the application of a negative bias is actually a potentiation process, in which the conductance increases if measured under a positive bias, which is analogous to LTP plasticity. Actually, the variation in the conductance (Δ$G$) stimulated by a single pulse depends on the initial conductance of the e-synapse. The Δ$G$–$G$ switching statistics during depression are presented in Fig. 2b. Interestingly, according to the measured Δ$G$ stimulated by a single pulse or the measured $G$, we can mine historical information of the voltage applied across the e-synapse. Especially, the number of positive pulses applied to the e-synapse can be approximately calculated based on the measured values of Δ$G$ and $G$, as shown in Fig. 2c. For example, for $G = 20$ nS, the number of applied positive pulses is between two and three, while for Δ$G = −5$ nS, the number of applied positive pulses is between four and six. This characteristic of our e-synapse is potentially useful for recovering critical information in the memory system. One should also note

that the width of the applied pulse can be further decreased. However, the voltage amplitude and the number of pulses need to be increased accordingly in order to achieve an appreciable change in the device conductance (Supplementary Fig. 6).

Because of the good flexibility of pMSSQ, the e-synapses exhibit an excellent operation endurance in the context of pulse cycling, including potentiation-depression cycling stress tests before/after bending, as shown in Fig. 2d. The e-synapse in the flat state shows reliable potentiation–depression functions with clear current differences between the low-conductance and the high-conductance states. The potentiation–depression cycling stress test is conducted after the 3D-ASN has been bent into a curve with a radius of the surface curvature of 10 mm. The potentiation–depression cyclic performances are stable and reproducible, regardless of repeated mechanical bending. Furthermore, the 3D-ASN is bent with a radius of curvature of 10 mm for 500 cycles. Each bending cycle includes one compression and one extension of the functional film. No noticeable performance degradation is observed, indicating an outstanding mechanical deformation endurance of the 3D-ASN (Supplementary Fig. 7).

The energy consumption for one operation can be calculated by multiplying the pulse amplitude by the current flowing across the device at each point in time (d$E = V \times I \times$ d$t$) and taking an integral over the operating time. Especially, when a switching from the high-conductance to the low-conductance state is considered, the energy consumption is much higher than that from the low-conductance to the high-conductance state due to the high resetting current required, particularly, for filament-

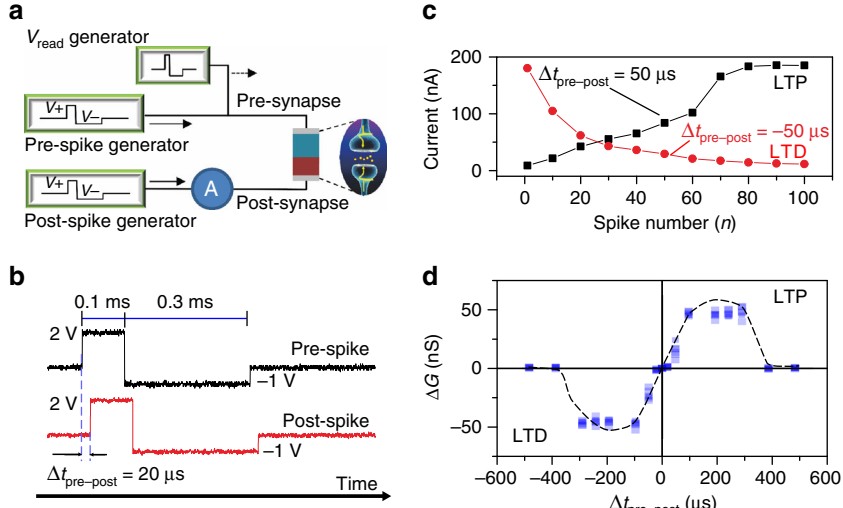

**Fig. 3** Correlated learning: spike-timing-dependent plasticity. **a** Electrical implementation for STDP measurements, in which an e-synapse is connected between spike generators, which act as neurons that send pre-synaptic and post-synaptic spikes to the e-synapse. After the applied pre-synaptic and post-synaptic spikes, a reading pulse is applied to detect the conductance of e-synapse. **b** Recorded waveform of the spikes applied to the e-synapse for the STDP demonstration. Each spike consists of a positive pulse (2 V, 0.1 ms), followed by a negative pulse (−1 V, 0.3 ms). $\Delta t_{pre-post}$ is defined as the interval from the initial time of the pre-synaptic spike to that of the post-synaptic spike. The value of $\Delta t_{pre-post}$ is positive (negative) when the pre-synaptic spike is applied before (after) the post-synaptic spike. **c** Current in the e-synapse as a function of the number of spikes for LTP and LTD stimulations performed with $\Delta t_{pre-post} = 50\,\mu s$ (LTP) and $\Delta t_{pre-post} = -50\,\mu s$ (LTD). **d** $\Delta G$ of the e-synapse stimulated by using a pair of spikes with different values of $\Delta t_{pre-post}$ ranging from −500 to 500 μs. When the pre-synaptic spike is applied shortly before (after) the post-synaptic spike, the synaptic weight increases (decreases), leading to the LTP (LTD). The *squares* represent the measured data, and the *dash line* indicates the trend curve

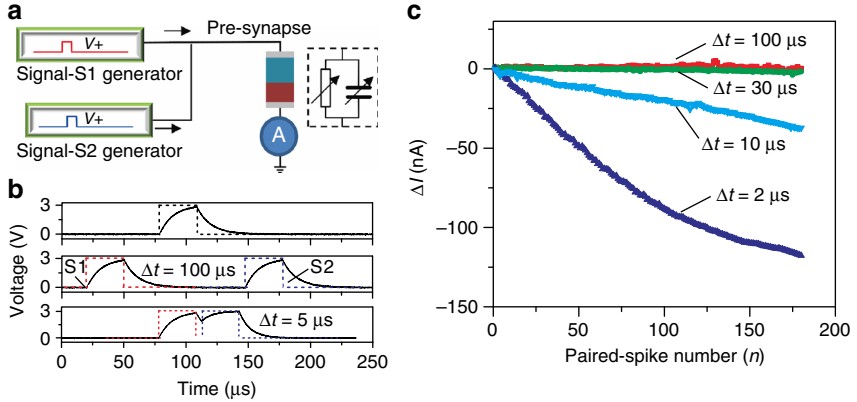

**Fig. 4** Correlated learning: paired-pulse facilitation. **a** Electrical implementation for paired-pulse facilitation measurements, in which an e-synapse equivalent to a parallel resistor–capacitor is connected to two spike generators, which act as neurons that send different pre-synaptic spikes to the e-synapse. **b** Recorded waveforms of the pulses applied to the e-synapse. The real spike (*solid lines*, 3 V, 30 μs) applied to the e-synapse is different from the square pulse (*dash lines*) generated from the signal generator due to a charge–discharge process in the resistor–capacitor loop (*top panel*). For the paired spikes with $\Delta t$ larger than ~50 μs, the spikes can be considered as independent stimulations to the device, and no significant variations of the synaptic strength can be observed (*middle panel*). As $\Delta t$ is decreased to 5 μs, the second pulse continues charging the resistor–capacitor loop (*bottom panel*). **c** Dependence of the current variation on the degree of relevancy and the number of paired spikes. As $\Delta t$ is decreased, a downward trend of the current with increasing number of paired spikes becomes manifest

based resistive switching devices. In this work, the highest current in the high-conductance state is <200 nA; thus, a low energy consumption is achieved. Figure 2e shows the energy consumption of a single e-synapse in both depression and potentiation processes. The higher the resistance of the initial state is, the lower the energy per pulse the device consumes. For the first pulse stimulation in the depression process, the energy consumption is about 60 pJ. As a result, the energy consumption also decreases with increasing number of pulse stimulations. The energy consumption is smaller than 0.5 pJ per pulse due to the low operating current.

**Correlated learning**. The synaptic strength in a biological synapse is modified by the dynamically and temporally correlated pre-synaptic and post-synaptic spikes via the STDP rule, which is one of the essential learning/memory laws for emulating synaptic functions[42]. LTP in canonical STDP occurs when pre-synaptic spikes lead post-synaptic spikes, and LTD happens when post-synaptic spikes lead pre-synaptic spikes[43]. For our e-synapses, the observed spike-rate-dependent characteristics are similar to those for the STDP rule of biological synapses. A pair of pulses ($V_+/V_-$ = 2 V, 0.1 ms/−1 V, 0.3 ms) are applied to the top and the bottom electrodes as pre-synaptic and post-synaptic spikes to implement

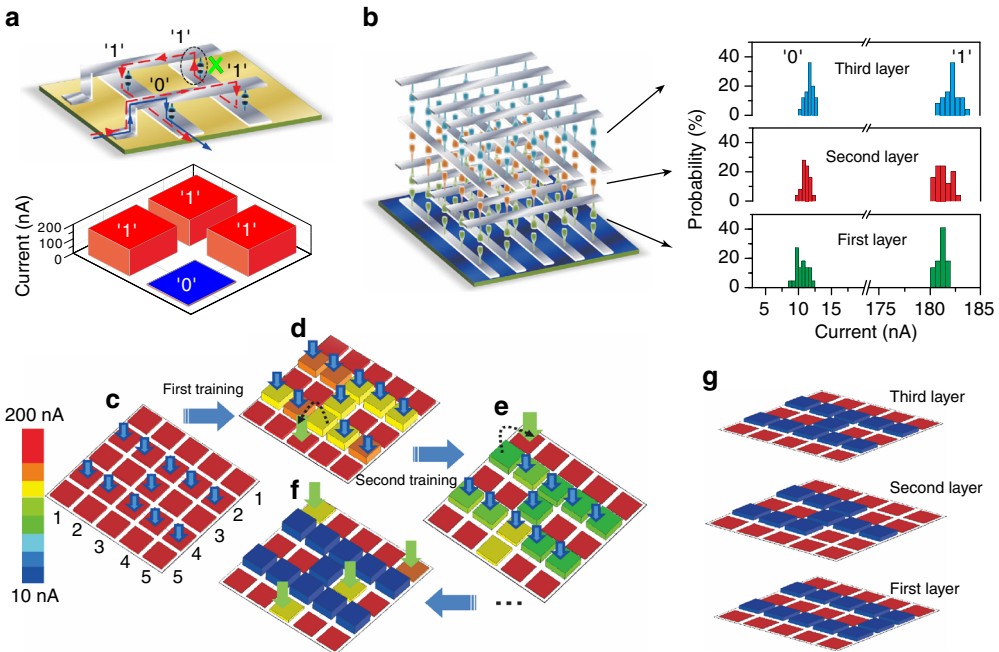

**Fig. 5** Trainable 3D memory performances. **a** Crosstalk effect existing in the memory array without selector devices (the *red dotted line*). In our 3D-ASN, the reverse current path can be blocked due to the rectification characteristic of the e-synapse. The *bottom panel* shows the readout current, which can correctly give the conductance of each cell. **b** Statistical distributions of the readout currents of all e-synapses in "0" and "1" states. **c**–**f** Image being memorized into the synapse array through a step-by-step training process. The figures correspond to the readout-current maps of the e-synapse array. **c** The initial readout-current states of the e-synapse array. The *blue arrows* are related to the input signals in the next training process. **d** The readout-current states after the first training process. The *blue arrows* correspond to the input signals in the next training process, in which an incorrect input signal (*green arrow*) is introduced. **e** The readout-current states after the second training process. The *blue arrows* correspond to the input signals in the next training process, in which an incorrect input signal (*green arrow*) is introduced. **f** The final readout-current states after 12 training processes, in which several interference e-synapses exist (*green arrows*). **g** The final readout-current maps of the three stacking layers after sufficient training

STDP, as illustrated in Fig. 3a, b. The relative timing $\Delta t_{\text{pre–post}}$ is defined as the interval from the initial time of the pre-synaptic spike to that of the post-synaptic spike. After the pre-synaptic and the post-synaptic spikes have been applied, a reading voltage (Supplementary Fig. 8) is applied to read the conductance of the e-synapse. The currents as functions of the number of spikes (spike number) measured in a STDP test with $\Delta t_{\text{pre–post}} = 50\,\mu s$ and $\Delta t_{\text{pre–post}} = -50\,\mu s$ are presented in Fig. 3c, respectively. When the pre-synaptic spikes are applied before the post-synaptic spikes ($\Delta t_{\text{pre–post}} = 50\,\mu s$), the current through the device gradually increases with increasing spike number, resulting in the achievement of LTP. In contrast, when the pre-synaptic spikes are applied after the post-synaptic spikes ($\Delta t_{\text{pre–post}} = -50\,\mu s$), LTD appears.

The $\Delta G$ of the e-synapse stimulated by a pair of spikes is measured under different values of $\Delta t_{\text{pre–post}}$ ranging from $-500$ to $500\,\mu s$, and the results are shown in Fig. 3d. When the pre-synaptic spike is applied shortly before (after) the post-synaptic spike, the synaptic weight is increased (decreased). In particular, when pairs of pre-synaptic and post-synaptic spikes are applied with a $|\Delta t_{\text{pre–post}}| > 400\,\mu s$, which means that the two input spikes have a low degree of degree of relevancy, the spikes can be considered as independent stimulations to the device, and no significant variations in the synaptic strength occur.

A more interesting phenomenon, similar to the learning-experience behavior of human beings, is also observed in our e-synapses. Pre-synaptic spikes from different neurons are well known to be able to trigger a post-synaptic current through synapses in a post-synaptic neuron to establish dynamic logic in a neural network[44]. PPF among the basic dynamic logic functions is demonstrated in our e-synapse by applying two successive

pre-synaptic spikes with an amplitude of 3 V, a width of 30 μs, and different inter-spike intervals ($\Delta t$), as shown in Fig. 4a. The signal generators emulate the production of pre-synaptic spikes from different pre-synaptic neurons. The correlated learning behavior could be explained on the basis of the equivalent circuit model and the electrical behavior of the e-synapse, with the e-synapse being equivalent to a resistor in parallel with a capacitor, as shown in Fig. 4a. The real spike applied to the e-synapse is different from the square pulse generated from the signal generator due to a charge–discharge process in the resistor–capacitor loop, as shown in the *top panel* of Fig. 4b. Furthermore, when the width of the generated square pulse is reduced sufficiently, the peak voltage applied to the e-synapse is smaller than 3 V, which could not effectively modulate the conductance of the e-synapse (Supplementary Fig. 9). For the paired spikes with $\Delta t$ larger than ∼50 μs, which means that the two input spikes have a low degree of relevancy, the spikes could be considered as independent stimulations to the device, and no significant change in the synaptic strength occurs, as shown in the *middle panel* of Fig. 4b. However, when $\Delta t$ is decreased sufficiently so that it is smaller than the fully-discharged time, the second pulse is able to keep charging the resistor-capacitor loop. Thus, the peak voltage applied to the e-synapse is able to reach 3 V, and the width of the applied spike also increases, as shown in the *bottom panel* of Fig. 4b. Physically, the overlap of the two pulses with a small $\Delta t$ accelerates ion migration, leading to a larger change in the conductivity.

For the demonstration of one of the dynamic logic functions, we define the degree of relevancy of the two signals as $\Delta t$. Thus, a smaller $\Delta t$ corresponds to a higher degree of relevancy. Because

we are interested in the variation of the weight of the e-synapse, we record the variation of the current following the paired spike. A clear dependence of the current variation on the degree of relevancy and the paired-spike number can be observed in Fig. 4c, with a high degree of relevancy being the most effective and a low degree of relevancy being the least effective. When two spikes (3 V, 30 μs) with a $\Delta t$ of 100 μs are applied, the conductance does not decrease, not even after 150 paired spikes have been applied. In other words, two signals with a low degree of relevancy are unable to change the synaptic weight, so our e-synapse is unable to learn the input signals. However, a downward trend in the current becomes manifest as the number of paired spikes is increased due to an increase in the degree of relevancy. This means that two signals with a high degree of degree of relevancy could be learned gradually and be remembered by our e-synapse. Moreover, a higher degree of relevancy should decrease the number of paired spikes necessary for achieving a target synaptic weight and speed up the correlated learning process.

**Trainable 3D-memory function**. The 3D-ASN employing our e-synapse exhibits a trainable memory function, as shown in Fig. 5. Typically, the parasitic paths (the *red dashed line* in Fig. 5a) for traditional crossbar memory devices without selector devices exist in parallel with the low-conductance (defined as "0" state) cell through the neighboring high-conductance (defined as "1" state) cells. As mentioned in the previous section, the crosstalk effect can be mitigated by combining the memory device with another functional device, usually called a selector. However, there has been no mature selector available at the moment. A proper utilization of our e-synapses can directly suppress the crosstalk without the need of any selector because the reverse current path (highlighted by the *dashed line circle* in Fig. 5a) can be blocked due to the rectification characteristic of our e-synapse. Current-addressing test measurements are executed in order to understand this effect. Initially, one synapse is set to the "0" state and its neighboring synapses are set to "1" states. The device shows precise addressing (the *blue solid line* in Fig. 5a) due to the rectification characteristic of the e-synapse. As a result, the information stored in the "0" state with a readout current of 8.7 nA could be perfectly detected (the *bottom panel* of Fig. 5a). Figure 5b shows the statistical distributions of the readout currents for all of the operative e-synapses in the "0" and the "1" states. The e-synapses in each layer of the 3D memory show a high uniformity in spite of its variation, which allows the storage of information in the 3D space.

Unlike the traditional digital memory devices, in which the data are stored by a single trigger of a setting or a resetting pulse, our 3D-ASN memorizes events and exercises cognition through repeated training processes. The memory of the human brain has strong tolerance for and robustness against input faults and variations. Thus, the realization of human-brain-like memorization plays an important role in artificial intelligence technology.

The memorization of an image by an e-synapse array is carried out to demonstrate the trainable-memory behaviors, as well as the strong tolerance and robustness, of the 3D-ASN, as shown in Fig. 5c–f. The synapse readout-current map is slightly modified toward the depression direction according to the learning algorithm used to mimic the input pattern. Initially, all of the e-synapses are set to the high-conductance state (Fig. 5c). The image of "H" is then programmed into the array to train the e-synapse array to remember the image. For each training session, a training pulse (3 V, 10 ms) is applied to the selected e-synapse, as indicated by the *blue arrows*. Note that only the selected layer is addressed and that the electrodes in other layers are floating in this programming process. After the first training process, the

resistance of the e-synapse begins to diverge, the conductance of the selected e-synapses is slightly suppressed, and the image of "H" starts to emerge on the readout-current map. To demonstrate the tolerance for and the robustness against incorrect input signals that can be introduced by electronic noises or timing errors, we artificially introduce an incorrect input signal in the second training process, as shown by the *arrows* in Fig. 5d. After the second training process, the conductance of the selected e-synapses are further decreased, indicative of a deeper memory of the "H" image. Note that the conductance of one undesired e-synapse is also decreased due to the incorrect input signal in the second training process (Fig. 5e). We randomly introduce some incorrect input signals into the ensuing training processes. The system is found to be insensitive to incorrect input signals, and many unintentional mistakes in the input signal can be accommodated by increasing the number of training processes. After the training with 15 "H" images and some random incorrect inputs, as shown in Fig. 5f, the final conductance of the selected e-synapse tends toward stability and easy identification of the learning target, and the input image of the "H" is strongly remembered, despite the existence of several interference e-synapses, as indicated by the *green arrows* in Fig. 5f. However, when such incorrect setting signals are input into a traditional digital memory device, the correct stored information is very difficult to recognize. Images of "Y" and "U" could be stored in the second and the third layers to realize 3D storage by using a similar training process, as shown in Fig. 5g.

## Discussion

The conductance change of our e-synapse is attributed to the migration of Cu ions induced by an external electric field. Worth noting is that the Cu ions can diffuse along both the lateral and the longitudinal directions due to the difference in the doping concentration of the Cu ions, and such diffusions could affect the long-term stability of the device. As shown in the stability measurements (Supplementary Fig. 5), the e-synapse exhibits LTP/LTD characteristics, indicating that neither the lateral nor the longitudinal diffusion has substantially affected the long-term stability of the e-synapse. However, when the thickness of the active layer and the lateral distance between cells are aggressively reduced further to obtain an ultra-high density integration, the effects of lateral and longitudinal diffusions of the Cu ions on the long-term stability may not be negligible. The ultra-low working current of our e-synapses is important for decreasing the energy consumption of the prospective high-density 3D synapse network. The working current of a single e-synapse partially depends on the active lateral size of the device. Therefore, scaling the lateral size down to submicrons or nanometers can decrease the working current of a single e-synapse, making an exact measurement of the working current difficult. Nevertheless, this issue can be addressed by thinning down the thickness of the active layer. In addition, the doping concentration and the type of doped ions can be engineered to maintain a reasonable working current in the range of nanoamperes.

Even though trainable memory behaviors were demonstrated by directly inputting images in this work, the presented high-performance memory effect of our 3D memory–system is important for further diverse neural-network applications because the memory effect obtained by changing the connection strength between neurons is a building block for advanced neural activity. The related algorithm and circuit[45, 46], the perceptron[25, 30], the visual system[23], the video-processing system, and the auditory processing system[13] can be further developed by combining the 3D-ASN with the existing artificial-neural-network model. Furthermore, by addressing two or three neighboring layers, the measured output currents are the arithmetic sum of all the

currents across the addressed e-synapses distributed in the addressed 3D space. In other words, various e-synapses can be regarded as having different weights corresponding to their conductances. Thus, the output signals are the arithmetic sum of all the weighted input signals[47]. In this case, our 3D-ASN provides a new physical platform for operating a complex algorithm for hierarchical neural networks[47].

In conclusion, we demonstrate a flexible 3D-ASN using e-synapses acted as memory units connected by 3D stacking crossbar electrodes. The 3D-ASN can successfully suppress the crosstalk without the need for any additional selector devices, which should decrease the complexity of the device structure and the fabrication process. Our e-synapse behaves as a biological synapse and exhibits the characteristics of unilateral connection, LTP/LTD, the STDP learning rule, PPF, and ultralow-power consumption (pJ range). The 3D-ASN exhibits the biological features, with the experimental evidence supporting correlated learning, a trainable memory, 3D storage, and strong fault-tolerant properties. The present observations provide good motivation for further study to envision a large-scale electronic system that could feasibly lead to rapid progress in manufacturing technology and artificial intelligence technology.

## Methods

**Preparation of Cu-doped pMSSQ**. pMSSQ was used as the matrix material. The pMSSQ was prepared by using methyltrimethoxysilane as a precursor as follows: A mixture of trimethoxymethylsilane (98%, Aldrich), n-butanol (99.8%, Sigma-Aldrich), and deionized water in a weight ratio of 4:10:1 was stirred at 60 °C for 24 h. Then, the mixture was aged for a week to increase its viscosity for achieving better film deposition coverage by using spin-coating. $CuCl_2$ powder (99.9999%, Sigma-Aldrich) was dissolved in deionized water to obtain a concentration of 50 mg/mL. For fabrication of the active layer of the e-synapse, we used a composite solution consisting of the methyltrimethoxysilane precursor, the copper-chloride aqueous solution, and methanol. Two kinds of Cu-doped methyltrimethoxysilane precursors were prepared. For the heavily Cu-doped methyltrimethoxysilane precursor, the methyltrimethoxysilane precursor, the $CuCl_2$ aqueous solution, methanol, and deionized water were mixed in a volume ratio of 50:10:5:10. For the lightly Cu-doped methyltrimethoxysilane precursor, the methyltrimethoxysilane precursor, $CuCl_2$ aqueous solution, methanol, and deionized water were mixed in a volume ratio of 50:1:5:19. Both mixtures were stirred at 60 °C for 24 h.

**Fabrication of 3D-ASN**. To fabricate the flexible 3D stacking devices, we initially cleaned polyethylene glycol terephthalate (PET) substrates in an ultrasonic bath with acetone, methanol, and deionized water in sequence for 30 min each. The bottom Al electrodes with six lines with widths of 1.5 mm each were deposited by using thermal evaporation with a shadow mask. The lightly Cu-doped methyltrimethoxysilane precursor was spin-coated onto the PET/Al substrate at 300 r.p.m. for 10 s and subsequently at 6000 r.p.m. for 30 s. The coated film was annealed on a hot plate at 90 °C for 10 min. After the baking, the heavily Cu-doped methyltrimethoxysilane precursor was spin-coated onto the PET/Al/Cu-PMSSQ sample at 300 r.p.m. for 10 s and subsequently at 6000 r.p.m. for 30 s. The coated film was baked at 90 °C for 10 min. Then, the contact pads of the bottom electrodes were exposed for the electrical measurements, and after that, the sample was hard-baked at 160 °C for 1 h to achieve complete cross-linking of the pMSSQ. The thickness of the Cu-pMSSQ layer was about 80 nm. The formation of the first active layer was followed by the deposition of other Al electrodes, whose directions were perpendicular to those of the bottom Al electrodes. The top Al electrodes of the first active layer were simultaneously the bottom electrodes of the second active layer. To form the second and the third active layers, we repeated the processes described above. The finished 3D stacking memory array consisted of three layers of memory, each containing a crossbar-type array.

**Characterization**. The electrical measurements were performed using a semiconductor analyzer system (4200, Keithley), an oscilloscope (TDS 2024 C, Tektronix) and a waveform generator (33220 A, Agilent) in an atmospheric environment at room temperature. In the electrical measurements, all of the bias voltages were applied to the top Al electrodes while keeping the bottom Al electrodes grounded. SEM image was obtained by using Nova Nano SEM 230 system. The absorption spectra of the films were characterized by using an UV–Vis spectrophotometer (Lambda 650 S, Perkin Elmer). The UPS measurements were performed in an analysis chamber using a He discharge lamp (XPS-Theta Probe, Thermo Fisher Scientific). The resolution of the measurements was 50 meV.

**Data availability**. The data that support the findings of this study are available from the corresponding author upon reasonable request.

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

## Acknowledgements

This research was supported by the Basic Science Research Program through the National Research Foundation of Korea funded by the Ministry of Education, Science and Technology (2016R1A2A1A05005502). C.W. was supported by the Korea Research Fellowship Program through the National Research Foundation of Korea funded by the Ministry of Science, ICT and Future Planning (2015H1D3A1062276).

## Author contributions

T.W.K. and C.W. conceived the project, and C.W. and T.W.K. designed and performed the experiments and collected the data. C.W., T.W.K., H.Y.C., D.B.S., and J.J.Y. analyzed and discussed the data. All authors discussed the results and contributed to the writing of the manuscript.

## Additional information

**Competing interests:** The authors declare no competing financial interests.

