## [Peer Review File · Nature Communications]

Reviewers' comments:

Reviewer #1 (Remarks to the Author):

In this manuscript, the authors demonstrated a flexible electronic device to mimic the neurons activities with layer-by-layer resistive memory devices as e-synapses. The work is marginal and it is not recommended to be published in Nature communications as too many doubts and questions appeared in my mind when I read this manuscript.

1. The matrix (pMSSQ) of the composite material used in this work has already been studied for several years and you can easily find it in some literatures (e.g. Solid State Communications 151 (2011) 297-300).
2. Most of the resistive devices are fabricated use Cu as electrode as it is critical in the resistive memory device to form conductive filament. However, CuCl₂ was used as the Cu ion sources in this work, but they baked their devices at 140 oC (usually 160 oC used in other papers, even so, the Cl ion still remains), then what about the Cl ion? No any information can be found in this manuscript, so I would like to raise a question that how the authors can eliminate the effects on Cl ions in their study.
3. Figure 2b, the unit of y-axis should be absolute value as negative current was detected in negative voltage region.
4. I don't see any difference between the structure of Al (bottom)/lightly-doped layer/heavily-doped layer/Al (top) and the other structure of Al (bottom)/heavily-doped layer/lightly doped layer/Al (top) (Line 163, Page 8) if you just turn one of them upside down, as the authors claimed they chose the materials because the can form layer-by-layer structure using spin-coating without re-dissolution (Line 87, Page 4). If so, the curves in Figure 3e should be mirror symmetry. And also, I think re-dissolution cannot be avoided as the author just heated the coated film at 90 oC (very less cross-linking reaction would occur). The authors may verify the morphology using AFM.

Reviewer #2 (Remarks to the Author):

When it comes down to reviewing the manuscript, originality and the level of development in the application of authors' from the previous works has to be considered. In this manuscript, the authors employed Cu-doped pMSSQ layer to manipulate Cu ions diffusion at the electrode interface to demonstrate 3D trainable nemomorphic memory that does not require additional functional device to reduce the crosstalk effect.

In material science point of view, a memory has been previously realized with metal gold nano particle layer inserted between pMSSQ layers. However, the active layer in this paper showed different type of doping, carrier movement, and layer structure to firmly construct e-synapses. In addition, the authors showed detail work of Cu ions movement in the pSMMQ media and demonstrate originality of their fine work.

On the application side this paper realized robust 3D flexible trainable memory device. Each unit's electrical performance was stable and homogenous while the device is bent or constructed in 3D structure. Also, crosstalk effect in 2D frame was not shown without additional circuitry. However, firstly it is still questionable that authors' 3D memory array is free from crosstalk in 3D frame when the device is working all three layers simultaneously. In Figure 5g, authors have shown the device working in the 3D structure array. I just want to clarify whether this device can be working all three layers simultaneously because our mind does not work in 2D frame where it can perceive multiple sensory or thoughts simultaneously. Also, it is intriguing to see whether each layer's operation causes any disturbance at the different layers. Secondly, it is highly desirable to mention thermal stability (for example, room temperature, 60 oC or higher) because the concentration distribution of the copper ion

in pSMMQ media could be influenced by the thermal energy. Also, there were some uncertainties in their sentences and figures that can be improved to enhance the readership as listed below.

Line 134: "the distribution of the concentration of elemental Cu with was analyzed" may change to "the concentration distribution of elemental Cu was analyzed."

Line 199 to Line 206: The format of the parameters in equations 1 to 3 should be consistent in whole manuscript.

Line 410: The numbers (6X6) of crossbar-type should be rechecked (5X5 crossbars appear in Figure 5)

Line 414: I believe that the device is composed of top/bottom Aluminum electrode whereas author mentioned "top Ag electrode." Clarification is needed.

Line 653: The numbers are overlapped at the x axis in Figure 2c and 2d.

Line 665: The y axis in Figure 5b should be defined.

Overall, I believe that this manuscript should be accepted under minor revision.

Reviewer #3 (Remarks to the Author):

The paper presents a flexible 3D quasi-brain memory network, in which e-synapses resemble the features of biological synapses, with unilateral connection, long-term potentiation, long-term depression, and ultralow-power consumption in the picojoule range. Even though the paper is well organized and is supported by experimental results, the material does not seem to be novel and significantly advance the knowledge in this area. Memristive synapses have been researched for the last few years by several groups and it is not clear why this one is significantly better. For example, ①T. Chang, et al. "Short-Term Memory to Long-Term Memory Transition in a Nanoscale Memristor," *Acs Nano*, 2011. ② S. Yu, et al. "A Low Energy Oxide-Based Electronic Synaptic Device for Neuromorphic Visual Systems with Tolerance to Device Variation," *Advanced Materials*, Mar 25 2013. ③ Y. Li, et al. "Associative Learning with Temporal Contiguity in a Memristive Circuit for Large-Scale Neuromorphic Networks," *Advanced Electronic Materials*, Aug 2015.

Below are some additional comments to improve the manuscript.

1. The author claimed that both LTP and LTD were implemented, actually these devices exhibit LTD behaviors under different voltage polarities.

2. The implementation of e-synapses is a function of the biphasic pulse signal. It is not clear if this special device could implement other functions of e-synapses (i.e. STDP, STP, LTP)?

3. Switching speed seems lack of competitive. The author need to optimize switching speed.

4. The dimension size of the e-synapse seems too large to integrate with Micro/nano-electronics technology, and the physical structure characteristics should been provided, and how to do devices alignment among the transparent flexible multilayers should also been clarified well.

Reviewers' comments:

Reviewer #1 (Remarks to the Author):

Authors responded all the questions raised by the reviewers, however I am still not convinced with the novelty of the work to publish in Nat. comm.

Reviewer #2 (Remarks to the Author):

The authors revised the manuscript to kindly settle the questions regarding crosstalk and thermal stability that I raised. Even though previous work regarding e-synapses based on pMSSQ exist, the integration level in 3D structure shows feasibility of futuristic device. I still recommend the publication in Nature Communications.

Reviewer #3 (Remarks to the Author):

The authors have given targeted answer to my previous questions, and I think the work of flexible 3D quasi-brain memory network in this manuscript is also improved. To be published in the journal, three points still should be clarified in the work.

1. Why choose pMSSQ as the functional material? What's the benefit compared to the metal oxides or other electrolyte which reported in the previous paper?
2. Could the cell work with higher speed pulse (~ns), which would gain much lower power consumption?
3. Please show the scaling and high-density integration array in the work, which will strengthen the significance of this study as the artificial synapse network.

Reviewer #4 (Remarks to the Author):

The results presented in this manuscript are interesting, albeit not really novel.

First, this work adds a new material (Cu-doped silsesquioxane derivative) to a long list of materials used to fabricate synapse-like devices. Basically, the proposed devices show synaptic-like behaviors (PPF, STDP,...) but it is not clear what are the performance improvements. The proposed mechanism (Cu ion diffusion between a low-doped and high-doped regions) is reminiscent of the oxygen vacancies migration in OxRAM. The authors propose that the change of the device resistance may be due to modification of the energy barrier at the electrode/Cu-MSSQ, resulting from Cu ion accumulation but they do not provide any values for these energy barriers. Thus, while plausible, this claim remains hypothetical.

The claim that this device exhibits both LTD (depression) and LTP (potentiation) is wrong, or at least not clear at all. In figure 2-a, the conductance is always decreasing with the number of applied pulses both for positive and negative voltages. This issue should be clarified.

From a technological point of view, this material allows the fabrication on flexible substrates. This is fine for certain class of possible applications (albeit not discussed). Nevertheless, the authors do not report convincing results that this material/device is technologically relevant. It is not sufficient to report only two measurements, one before and one after a bending test. Mechanical cyclability was

not reported. The Cu-MSSQ material is also not relevant for down-scaling, as required to implement neural networks with high-density of synaptic devices. The measured currents are below 200 nA for a device with an area of 1.5mm x 1.5mm. The current density is low, which make impossible the use of this material to implement synaptic-like device with a lateral size of few tens of nanometers.

I do not clearly understand why the rectification behavior of the reported device allows to avoid cross-talk. I suppose that the "state" of the cells in the crossbar (Fig. 5-a) are read with a reading pulse in the reverse regime, but this is not clearly stated in the paper. But, in that case, it is not clear from data in figure 1 that a synaptic behavior is present in the reverse regime.

In conclusion, I recommend not publishing this manuscript, which is probably more relevant to a more specialist journal.

Reviewer #5 (Remarks to the Author):

In this manuscript the authors demonstrate a flexible 3D crossbar architecture. This architecture consists of selector-less memory nodes, made with a hybrid pMSSQ-based polymer. Several neuromorphic functions were demonstrated with these devices including STDP, PPF. Although the matrix material has been also used at past, from technological point of view the demonstration of a flexible 3D cross-bar architecture is still an advancement. The key feature is the selector-less memory nodes which ensure the elimination of the sneak paths from device-to-device and from layer-to-layer. I believe that the significance of the paper is more from technological point of view (i.e., realization of complex 3D architectures), than from the point of view of neuromorphic computing. In my opinion, the concept of the manuscript fits to the general scope of Nat. Comm., but there are many points/inconsistencies that have to be revised and written more carefully.

1. At the end of the abstract the authors claim that the proposed 3D geometry offers memory capability comparable to biological neural networks. At lines 50-57, p. 5 the authors use similar arguments about 3D architectures. 3D interconnectivity with cross-bar architectures can indeed offer multidimensional topologies, but it is limited to regular topologies such as multilayer nets (vector-matrix multiplication schemes in cross-bars, etc). There is a whole zoo of biological/arbitrary networks which cannot be physically realized with the cross-bar architecture. The authors could use here more realistic arguments.

2. At lines 66-68, p. 4, the authors refer that "Even though synaptic plasticity has been extensively demonstrated, the use of one inherent characteristic of a chemical synapse, the one-direction transmission of signals, has rarely been reported". The authors could explain in more detail the "one-direction transmission of signals" and its implications in the neural environment.

3. At lines 92-93, p. 5, the authors refer that the dominant mechanism of the resistive switching is "still a subject of debate, the movement of the cations and their interactions are the dominant mechanisms". The authors could be more specific here regarding the switching mechanisms.

4. At line 113, p. 6 there is a mistake. The authors refer to "Fig. 1e" instead of "Fig. 1d".

5. At line 133, p. 7 the authors use the term "adaptation" for describing the modulation of the synaptic weight. This might create confusion, because adaptation is a terminology used for another biological/behavioral process.

6. At Fig. 2a, the current resolution does not seem to be enough for the measurements. Also, at Fig. 2a, the device exhibits only LTD and not LTP for positive and negative pulses. Therefore, the authors should carefully revise lines 138-143. Also, at line 143, p. 7, the authors use the term "plastic rate". The authors should revise the term because it might lead to misunderstandings.

7. At Fig. 3, the authors demonstrate STDP with overlapping pulses. What happens when using non-overlapping pulses?

8. At Fig. 5, the authors demonstrate that the elements of each layer can be programmed independently. Are there any sneak paths across different layers? If not, is this beneficial for neuromorphic architectures? In the case of "isolated" layers, how can the information flow in 3D space be achieved?

REVIEWERS' COMMENTS:

Reviewer #3 (Remarks to the Author):

Authors responded all my questions and make necessary supplement in the revised manuscript. I think it could be accepted by Nature Communications now.

Reviewer #4 (Remarks to the Author):

The authors have correctly responded to my questions and comments. The paper has been significantly improved. It is suitable for publication in Nature Communications.

Reviewer #5 (Remarks to the Author):

The authors addressed adequately most of my comments. The manuscript has now reached a satisfactory level for publication in Nat. Commun. However, a few questions rise through the revision process and the authors may want to address them in order to further improve the quality of the manuscript, or even stimulate future directions. More specifically:

1. Have the authors studied the long-term lateral Ca^{2+} diffusion from cell to cell through stress measurements? This might be important for the long-term stability of their architecture.
2. A slightly better explanation is needed for the layer to layer communication. At the revision of p. 17, the authors refer that "by addressing two or three neighboring layers, the output signals are the sum of all the input signals across all the addressed e-synapses distributed in the addressed 3D space." Regarding Figs c-g, if I understand correctly, by addressing three cells from neighboring layers (for example cell (1,4) from layers 1 and 2 and 3), the output will be overall (arithmetic) sum? The authors could clarify this delicate point.

Reviewers' comments:

Reviewer #1 (Remarks to the Author):

In this manuscript, the authors demonstrated a flexible electronic device to mimic the neurons' activities with layer-by-layer resistive memory devices as e-synapses. The work is marginal and it is not recommended to be published in Nature Communications as too many doubts and questions appeared in my mind when I read this manuscript.

1. The matrix (pMSSQ) of the composite material used in this work has already been studied for several years and you can easily find it in some literature (e.g., Solid State Communications 151 (2011) 297-300).

Response:

Thank you for this important comment. We are certain that the reviewer realizes that much innovative research is based on previously-used materials. Although it is true that pMSSQ has been studied for several years, our research used that material to investigate areas not previously investigated and produced results not previously reported. As a result, this study contains significant innovative content on the application of pMSSQ in artificial synapses. Further, we believe the demonstration of these significant innovations is not significantly affected by the fact that this material was used in previous research.

2. Most of the resistive devices are fabricated using Cu as an electrode as it is critical in the resistive memory device to form conductive filaments. However, CuCl₂ was used as the Cu-ion source in this work, but they baked their devices at 140°C (usually 160°C used in other papers; even so, the Cl ion still remains); then, what about the Cl ion? No information can be found in this manuscript, so I would like to raise a question about how the authors can eliminate the effects on Cl ions in their study.

Response:

Thank you for this important comment. We totally agree that the redox reaction of Cu ions is the working mechanism of most Cu-filament-based resistive memory devices. However, in our work, we have demonstrated that the energy gap of the Cu-pMSSQ decreased with increasing copper-ion concentration and that the excellent insulating pMSSQ layer could be transformed into a semiconducting layer by doping with a high concentration of copper ions, which is totally different from the working mechanism of metal-filament-based resistive memory devices. As suggested by the referee, we have tried to verify the effects of Cl ions by adding HCl or NaCl into pure pMSSQ, and similar synaptic behaviors were not observed. Therefore, we definitely believe that the Cu ions play the key role in the memristive behavior.

3. Figure 2b, the unit of the y-axis should be an absolute value as a negative current was detected in the negative voltage region.

Response:

Thank you for bringing this to our attention. We have rearranged the structure of our manuscript and revised the figures.

4. I don't see any difference between the structure of Al (bottom)/lightly-doped layer/heavily-doped layer/Al (top) and the other structure of Al (bottom)/heavily-doped layer/lightly doped layer/Al (top) (Line 163, Page 8) if you just turn one of them upside down, as the authors claimed they chose the materials because they can form a layer-by-layer structure using spin-coating without re-dissolution (Line 87, Page 4). If so, the curves in Figure 3e should be mirror symmetry. Also, I think re-dissolution cannot be avoided as the author just heated the coated film at 90°C (very little cross-linking reaction would occur). The authors may verify the morphology using AFM.

Response:

Thank you for this important comment. In our original manuscript, we fabricated two devices, one with an Al (bottom)/lightly-doped layer/heavily-doped layer/Al (top) structure and the other with an Al (bottom)/heavily-doped layer/lightly doped layer/Al (top) structure, to show that the electrical properties of our devices could be modulated by using the active layer structure. In the revised manuscript, we have focused on the performance and potential applications of the 3D artificial synapse network. Furthermore, in the revised manuscript, we have added exciting results, provided a relevant discussion about the device's performance, and removed some of the materials-related discussion.

Reviewer #2 (Remarks to the Author):

When it comes down to reviewing the manuscript, originality and the level of development in the application of the authors' from the previous works has to be considered. In this manuscript, the authors employed a Cu-doped pMSSQ layer to manipulate Cu-ion diffusion at the electrode interface to demonstrate 3D trainable neuromorphic memory that does not require an additional functional device to reduce the crosstalk effect.

In a material science point of view, a memory has been previously realized with a metal gold nano particle layer inserted between pMSSQ layers. However, the active layer in this paper showed a different type of doping, carrier movement, and layer structure to firmly construct e-synapses. In addition, the authors showed detailed work on the Cu-ions' movement in the pSMMQ media and demonstrate originality of their fine work.

On the application side, this paper realized a robust 3D flexible trainable memory device. Each unit's electrical performance was stable and homogenous while the device is bent or constructed in a 3D structure. Also, the crosstalk effect in the 2D frame was not shown without additional circuitry. However, firstly it is still questionable that authors' 3D memory array is free from crosstalk in the 3D frame when the device is working all three layers simultaneously. In Figure 5g, the authors have shown the device working in the 3D structure array. I just want to clarify whether this device can be working all three layers simultaneously because our mind does not work in a 2D frame where it can perceive multiple sensory or thoughts simultaneously. Also, it is intriguing to see whether each layer's operation causes any disturbance at the different layers. Secondly, it is highly desirable to mention thermal stability (for example, room temperature, 60°C or higher) because the concentration distribution of the copper ion in the pSMMQ medium could be influenced by the thermal energy. Also, there were some uncertainties in their sentences and figures that can be improved to enhance the readership as listed below.

Response:

Thank you for your comment concerning the originality of our work and for your important observations. Our 3D artificial synapse network is free of the cross-talk problem in a 3D

frame when all three layers in the device are simultaneously working. Therefore, we can successfully obtain the statistical distributions of the readout currents for all of the operative e-synapses in the “0” and the “1” states, as shown in Fig. 5b. The information can be simultaneously stored in the three layers, as shown in Fig. 5g. We totally agree with the thermal stability question raised by the reviewer. Although it is true that the concentration distribution of the copper ion can be significantly affected by the thermal energy, we believe that, due to the low working current of our devices, the temperature can be easily controlled in practical applications by adopting an external cooling system.

Line 134: “the distribution of the concentration of elemental Cu with was analyzed” may change to “the concentration distribution of elemental Cu was analyzed.”

Line 199 to Line 206: The format of the parameters in equations 1 to 3 should be consistent in the whole manuscript.

Line 410: The numbers (6X6) of crossbar-type should be rechecked (5X5 crossbars appear in Figure 5)

Line 414: I believe that the device is composed of top/bottom aluminum electrode whereas author mentioned “top Ag electrode.” Clarification is needed.

Line 653: The numbers are overlapped at the x axis in Figure 2c and 2d.

Line 665: The y axis in Figure 5b should be defined.

Response:

Thank you for these important corrections. We have rearranged the manuscript and revised the figures and ambiguous sentences as you indicated.

Overall, I believe that this manuscript should be accepted under minor revision.

Reviewer #3 (Remarks to the Author):

The paper presents a flexible 3D quasi-brain memory network, in which e-synapses resemble the features of biological synapses, with unilateral connection, long-term potentiation, long-term depression, and ultralow-power consumption in the picojoule range. Even though the paper is well organized and is supported by experimental results, the material does not seem to be novel and significantly advance the knowledge in this area.

Response:

Thank you for your important comment. Although we understand the reviewer’s comment, we would like to point out that much innovative research is based on materials used in previous research, as is the case with our research. Although pMSSQ has been studied for several years, we have used this material for innovative research on electronic synapses and have produced observations and results that have not previously been published. Furthermore, our research was the first attempt to apply pMSSQ in artificial synapses. Therefore, we definitely believe that the innovative results and conclusions presented in our work are not affected by that fact that the material we used was also used in many previous research efforts.

Memristive synapses have been researched for the last few years by several groups and it is not clear why this one is significantly better. For example, □ T. Chang, et al. "Short-Term Memory to Long-Term Memory Transition in a Nanoscale Memristor," Acs Nano, 2011. □ S. Yu, et al. "A Low Energy Oxide-Based Electronic Synaptic Device for Neuromorphic Visual Systems with Tolerance to Device Variation," Advanced Materials, Mar 25 2013, □ Y. Li, et

al. "Associative Learning with Temporal Contiguity in a Memristive Circuit for Large-Scale Neuromorphic Networks," Advanced Electronic Materials, Aug 2015.

Response:

Thank you for this important comment. Although it is true that synapses have been researched for the last few years by several groups, most of the research focuses only on the mimicking of various synaptic operations in a single device and a flat-panel array. Studies about the 3D interconnectivity of electronic e-synapses, which plays a vital role in high-density information storage and multi-dimensional information processing in biological neural networks, have not yet been conducted with existing electronic devices. This is an obstacle to the practical applications of artificial neural-network devices. The main focus of our work, which is different from those of previous works, is the demonstration of a flexible, 3D, stacked, artificial synapse network utilizing switch-device-free electronic synapses. We have given an extended discussion of the significance of our work in the Introduction section above.

Below are some additional comments to improve the manuscript.

1. The author claimed that both LTP and LTD were implemented; actually these devices exhibit LTD behaviors under different voltage polarities.

Response:

Thank you for this observation/question. The devices exhibit LTD behaviors under positive bias voltages and show LTP behaviors under negative bias voltages, as shown in Fig. 2a.

2. The implementation of e-synapses is a function of the biphasic pulse signal. It is not clear if this special device could implement other functions of e-synapses (i.e., STDP, STP, LTP)?

Response:

Thank you for this important concern. In the revised manuscript, the emulation of LTP, STDP, and paired-pulse facilitation by our devices have been demonstrated and discussed.

3. Switching speed seems not to be competitive. The authors need to optimize the switching speed.

Response:

Thank you for this important suggestion. Even though the switching time is between μs and ms , which is not short enough, as shown in Fig. 4, the switching speed of our e-synapse is almost the same as that of an actual biological synapse. Therefore, the emulation performances of our device are comparable to those of biological synapses.

4. The dimension size of the e-synapse seems too large to integrate with micro/nano-electronics technology, the physical structure characteristics should be provided, and how to do device alignment among the transparent flexible multilayers should be clarified.

Response:

Thank you for these important suggestions. In this work, we have focused only on the novel electrical performance of our 3D artificial synapse network with a high density, and we have presented a prototype device. The high-density integration and the alignment of the structure of the artificial synapse network will be achieved by using advanced semiconductor technology in a future work.

We are very grateful for the helpful comments, suggestions, and advice on our paper given by the senior editor and the reviewers, and we hope you will find that our newly-revised paper satisfies the requirements and is ready to be reconsidered for publication in *Nature Communications*. However, if in the opinion of the reviewers and the editor, further revisions and/or more details are needed prior to publication, we would appreciate your additional helpful comments. Thank you for your consideration.

With best regards,

Tae Whan Kim

HYU Distinguished Professor, Hanyang University

Department of Electronics and Computer Engineering

Tel: +82-2- 220-0354

Fax: +82-2-2292-4135

E-mail: twk@hanyang.ac.kr

Point-by-point responses to reviewers' comments (comments in italics, responses in blue, and content corrections in yellow)

Reviewers' comments:

Reviewer #1 (Remarks to the Author):

Authors responded all the requested raised by the reviewers, however I am still not convinced with the novelty of the work to publish in Nat. comm.

Response: We thank the reviewer for offering his/her comment and would like to emphasize the novelty of our devices: (1) they are experimental demonstration of three-dimensional and functional circuits; (2) they have great mechanical flexibility; (3) they exhibit inherent rectifying I-V characteristics for suppression of crosstalk effects, *i.e.*, one-direction transmission of signals; (4) they have small operation currents, leading to an ultralow power consumption necessary for high-density integration; (5) they have exhibited correlated learning and trainable memory capability with strong tolerances to input faults and variations. **To the best of our knowledge, no other device/circuit reported in the literature has all of these highly desirable characteristics only found in the bio-intelligent system, i.e., the brain.** We believe the significant contribution of our work to the development of neuromorphic computing or artificial intelligent lies in the fact that this study is **the first experimental demonstration of flexible, 3D stackable, artificial synapse networks that can be used in futuristic electronic devices and will provide a new physical platform for the realization of smart memory and machine learning and for the operation of the complex algorithms involved in hierarchical neural networks.** We, therefore, believe that the novelty of this work is sufficient for publication in *Nature Communications*.

Reviewer #2 (Remarks to the Author):

The authors revised the manuscript to kindly settle the questions regarding crosstalk and thermal stability that I raised. Even though previous work regarding e-synapses based on pMSSQ exist, the integration level in 3D structure shows feasibility of futuristic device. I still recommend the publication in Nature Communications.

Response: We thank the reviewer for acknowledging the originality and significance of our work.

Reviewer #3 (Remarks to the Author):

The authors have given targeted answer to my previous questions, and I think the work of flexible 3D quasi-brain memory network in this manuscript is also improved. To be published in the journal, three points still should be clarified in the work.

Response: Thank you for the positive comment. We will address the new questions in detail below.

1. Why choose pMSSQ as the functional material? What's the benefit compared to the metal oxides or other electrolyte which reported in the previous paper?

Response: This is a great question, which has helped us to further clarify an important point in the revised manuscript. For the design and fabrication of a flexible, 3D, artificial synapse network, several properties of the functional materials, besides mimicking synaptic behaviors, should be considered: (1) The functional materials must contain excellent flexibility. As a

result, the metal-oxide-based or inorganic-material-based functional materials (*Nature materials*, 16, 2017, 101-108; *Nature*, 521, 2015, 61-64; *ACS nano*, 8, 2014, 6998-7004.) are not the best choices for flexible circuits. (2) The functional materials should have good chemical and physical stabilities because the deposition process of the upper functional layer might degrade the lower functional materials that are deposited first. However, for most of the reported organic electrolytes or active materials (*Nature Materials*, 16, 2017, 414-418), the deposition of the upper functional layer by using a solution method might re-dissolve the previously deposited lower layers.

In our work, we used pMSSQ as the functional material because pMSSQ is an inorganic/organic, silicone-containing hybrid material. It has excellent mechanical properties that make it suitable for flexible electronics. Moreover, the PMSSQ layer can be fully cross-linked by thermal annealing at a relatively low temperature, which allows it to maintain good chemical and physical properties.

According to the comment, we have added some descriptions in the revised manuscript to help readers understand the advantage of pMSSQ. That description is as follows: “Furthermore, for typical organic materials, the lower functional layers might be partially re-dissolved during the deposition process or the lithography processes of the upper layers, which makes the preparation of a multilayered stacking structure very challenging. On the other hand, the utilization of inorganic functional layers is also an impediment to the realization of high-performance flexibility.” (Paragraph 1, Page 4)

“Furthermore, the matrix material used in this work, namely poly(methylsilsequioxane) (pMSSQ), has excellent flexibility, as well as good thermal, chemical, and physical stabilities, for use in flexible electronics³⁴. The fully cross-linked structure of pMSSQ allows the multilayered stacking structure to be fabricated layer-by-layer through a solution processable

approach without the problem of re-dissolving the previously deposited lower layers.”

(Paragraph 2, Page 4)

2. *Could the cell work with higher speed pulse (~ns), which would gain much lower power consumption?*

Response: As suggested by the reviewer, we have experimentally shown that the width of the applied pulse can be further reduced. However, the voltage amplitude and the number of pulse need be increased accordingly in order to obtain an appreciable change in the device conductance. Related information has been added in the Supporting Information. When we decreased the pulse width to 100 ns and increased the amplitude of the pulse to 10 V, as shown in Supplementary Fig. S8, approximately 1600 pulses were needed to achieve an detectable conductance change.

Even though the devices can be operated with a higher speed pulse, considering the width of biological synaptic spikes (~ms) and the full simulation of correlated learning, we have used pulses with widths in the range of milliseconds in the following discussion. According to

our answer to the reviewer's helpful question, we have added some sentences in the revised manuscript as follows: "One should also note that the width of the applied pulse can be further decreased. However, the voltage amplitude and the number of pulses need to be increased accordingly in order to achieve an appreciable change in the device conductance (see Supplementary Fig. S8)." (Paragraph 1, Page 9)

3. Please show the scaling and high-density integration array in the work, which will strengthen the significant of this study as the artificial synapse network.

Response: As the reviewer mentioned, we would like to acknowledge that the high-density integration array is definitely worth studying, and the results of such an effort may further strengthen the significance of this study. However, as the title of our manuscript indicates, this work focuses on demonstrating the possibility of realizing flexible, 3D, artificial synapse networks utilizing Cu-doped pMSSQ as the functional material. Therefore, to make this work more focused on the functionality demonstration, rather than the optimization of the process conditions for electrodes, we concentrated on the preparation of patterned electrodes with a shadow mask to make prototype devices. The crossbar electrodes should be scaled down to micron or nanometer scale in order to prepare a high-density integration array, which will need photolithography to prepare the patterned metal electrodes. However, the wet etching or the dry etching process might affect the morphology and the properties of the functional layer, detailed studies of these effects might divert the readers' attention from the main target of this work, the demonstration of a prototype device. For that reason, in this study, we prepared the patterned electrodes with a shadow mask only. We believe that the size of the devices does not significantly affect the main concept, results, and conclusion of our current work and that investigations about high-density integration serve as a great future work. Once again, we

thank the reviewer for pointing this out and agree that studies on high-density integration is definitely meaningful and important in understanding and applying flexible 3D, artificial synapse networks.

Reviewer #4 (Remarks to the Author):

The results presented in this manuscript are interesting, albeit not really novel.

Response: We thank the reviewer for offering his/her comment and would like to emphasize the novelty of our devices: (1) they are experimental demonstration of three-dimensional and functional circuits; (2) they have great mechanical flexibility; (3) they exhibit inherent rectifying I-V characteristics for suppression of crosstalk effects, *i.e.*, *one-direction transmission of signals*; (4) they have small operation currents, leading to an ultralow power consumption necessary for high-density integration; (5) they have exhibited correlated learning and trainable memory capability with strong tolerances to input faults and variations. **To the best of our knowledge, no other device/circuit reported in the literature has all of these highly desirable characteristics only found in the bio-intelligent system, i.e., the brain.** We believe the significant contribution of our work to the development of neuromorphic computing or artificial intelligent lies in the fact that this study is **the first experimental demonstration of flexible, 3D stackable, artificial synapse networks that can be used in futuristic electronic devices and will provide a new physical platform for the realization of smart memory and machine learning and for the operation of the complex algorithms involved in hierarchical neural networks.** We, therefore, believe that the novelty of this work is sufficient for publication in *Nature Communications*. We have done our best to describe these points in our responses to the following comments:

First, this work adds a new material (Cu-doped silsesquioxane derivative) to a long list of materials used to fabricate synaptique-like devices. Basically, the proposed devices show synaptic-like behaviors (PPF, STDP,...) but it is not clear what are the performance improvements.

Response: we thank the reviewer for sharing his/her concerns, which give us an opportunity to clarify some important points. The advantages of our new material for synaptic devices with improved performance are as follows:

(1) Flexible, 3D, stacking devices achieved by utilizing our Cu-doped pMSSQ are in fact fairly challenging with other type of materials on the long list of synaptic materials. Typically, organic materials can be used to fabricate flexible synaptic devices, such as organic mixed ionic/electronic conductors or organic/inorganic nanocomposites. However, when a multilayered stacking structure is prepared, the lower layers deposited earlier might be chemically or physically destroyed during the deposition process of the upper layers deposited latter. With the use of inorganic-based materials, such as metallic oxides and metal sulfides, a multilayered stacking structure can be fabricated without much problem. However, achieving a high-performance device that is at the same time flexible is impossible due to the fragility of metallic-oxide or the metal-sulfide film. The pMSSQ has excellent flexibility, as well as excellent thermal, chemical, and physical stabilities, and is ideal for use in the fabrication of flexible electronics. In particular, the fully cross-linked structure of pMSSQ allows the multilayered stacking structure to be fabricated layer-by-layer by using a solution method without re-dissolving the lower layers. Therefore, we do believe that it is the best choice so far for the fabrication of flexible 3D devices.

(2) The multilayered stacking architecture with a crossbar structure is an excellent candidate for achieving 3D interconnectivity. However, the sneak path current issue is a well-known problem that prevents a normal crossbar array from being electrically operated

appropriately. As a possible solution, the electronic component at each crosspoint is connected to a selector device to suppress the sneak path current effect. However, there has been no idea selector available so far. In addition, this might decrease the integration level of the array and increase the complexity of 3D interconnectivity. Our devices based on the Cu-doped pMSSQ have intrinsic rectification characteristics (one-direction transmission characteristics), which suppress the crosstalk effect naturally without using any selector devices, providing a simpler and inexpensive approach to achieve high-density integration.

(3) A pure pMSSQ is a good insulator with low leakage currents. Thus, the working current of a single device cell based on Cu-doped pMSSQ is ultralow, which is of importance to decrease the total power consumption required for potential high-density integration.

(4) Correlated learning and trainable memory effects with strong tolerances to input faults and variations have been demonstrated, which can provide a new physical platform for smart memories and machine learning.

On the basis of the above comments, we have rearranged the Introduction, as shown on pages 3 and 4, to highlight the advantages of this functional material.

The proposed mechanism (Cu ion diffusion between a low-doped and high-doped regions) is reminiscent of the oxygen vacancies migration in OxRAM. The authors propose that the change of the device resistance may be due to modification of the energy barrier at the electrode/Cu-MSSQ, resulting from Cu ion accumulation but they do not provide any values for these energy barriers. Thus, while plausible, this claim remains hypothetical.

Response: We appreciate the helpful suggestion about characterization of the Cu-doped pMSSQ. In supplementary Fig. S2, we have shown that the electrical conductivity of the Cu-doped pMSSQ film increases with increasing copper-ion concentration. In supplementary Fig. S3, we have shown that the energy gap of Cu-doped pMSSQ decreases with increasing

copper-ion concentration. Furthermore, the energy band structures of the pure pMSSQ and the Cu-doped pMSSQ have been analyzed by using ultraviolet photoelectron spectroscopy (UPS), according to the reviewer's valuable suggestion. The doping of Cu ions shifts the HOMO of Cu-doped pMSSQ to lower binding energy, as shown in supplementary Fig. S4(b), which can reduce the barrier to hole injection from the electrode to the functional material. The related contents have been described in the Methods and in supplementary Fig. S4 of the Supporting Information. Supplementary Fig. S4 for the UPS measurement is presented below:

The claim that this device exhibits both LTD (depression) and LTP (potentiation) is wrong, or at least not clear at all. In figure 2-a, the conductance is always decreasing with the number of applied pulses both for positive and negative voltages. This issue should be clarified.

Response: we thank the reviewer for pointing this out, which is indeed an easy source of confusion. The performance of our device is different from most of those reported devices, in

which a positive bias leads to potentiation (or depression), and a negative bias leads to depression (or potentiation). In our work, our concern is the conductance of the devices under **positive** bias because of the rectification characteristics of our e-synapses. Since the conductance under negative bias in our devices is always negligible compared with that under positive bias (due to rectification), the synaptic weight should be measured under positive bias. The conductance gradually decreases with increasing number of **positive** pulses, which corresponds to the **depression** process. It is true that the conductance measured under a **negative** bias also decreases with increasing number of **negative** pulses. After the application of enough **negative** pulses, if a **positive** bias is applied to measure the conductance state, and the device is found to have already returned to its original high-conductance state. This means that the application of **negative** biases is actually a potentiation process, in which an increase in conductance is induced if measured under a **positive** bias. We have rephrased the words to make the sentences clearer as follows: “It is worth noting that the conductance of the e-synapse should be measured under a positive bias because of its remarkable rectification characteristics and negligible conductance under a negative bias. After the application of sufficient negative pulses, the e-synapse returns to its original high-conductance state measured under positive pulses. This means that the application of a negative bias is actually a potentiation process, in which the conductance increases if measured under a positive bias, which is analogous to LTP plasticity.” (Paragraph 2, Page 8)

From a technological point of view, this material allows the fabrication on flexible substrates. This is fine for certain class of possible applications (albeit not discussed).

Response: Thank you for the helpful comment. The PMSSQ layer can be fully cross-linked by thermal annealing at a relatively low temperature, which is compatible with plastic

substrates for fabrication of flexible circuits. In the revised manuscript, we have described the experimental results about the flexibility test in more detail. We have added the following sentence in the revised manuscript: “Furthermore, the matrix material used in this work, namely poly(methylsilsesquioxane) (pMSSQ), has excellent flexibility, as well as good thermal, chemical, and physical stabilities, for use in flexible electronics³⁴.” (Paragraph 2, Page 4)

Nevertheless, the authors do not report convincing results that this material/device is technologically relevant. It is not sufficient to report only two measurements, one before and one after a bending test. Mechanical cyclability was not reported.

Response: we thank the review for the helpful comment and request. The stability of the e-synapse over 500 mechanical bending cycles has now been demonstrated and is presented in the supplementary Fig. S9, which is presented below:

The e-synapse shows outstanding mechanical deformation endurance. After every 10 bending cycles (radius of curvature: 10 mm) under positive and negative pulses, the electrical properties of the e-synapse were measured. The currents of the e-synapse at the 1st positive voltage pulse and the 11th positive voltage pulse are recorded as low- and high-resistance

states, respectively. The e-synapse still shows stable and repeatable potentiation-depression cyclic performances. The related content has been updated in the Supporting Information. The related description has been updated in the main text as follows: “Furthermore, the e-synapse showed outstanding mechanical deformation endurance (see Supplementary Fig. S9).” (Paragraph 2, Page 9)

The Cu-MSSQ material is also not relevant for down-scaling, as required to implement neural networks with high-density of synaptic devices. The measured currents are below 200 nA for a device with an area of 1.5mm x 1.5mm. The current density is low, which make impossible the use of this material to implement synaptic-like device with a lateral size of few tens of nanometers.

Response: We thank the reviewer for commenting on this point. The working current of a single e-synapse is an important factor, which is worth considering when it comes to high-density implementation. When the lateral size of a single e-synapse is scaled down to submicrons or nanometers, the measured current can be maintained by thinning down the thickness of the active layer. Moreover, the doping concentration and the type of doped ions can be further modified when the single e-synapse is scaled down. At the current stage of our research, we have focused on demonstrating the possibility of realizing flexible, 3D, artificial synapse networks by using a new functional material (Cu-doped pMSSQ) that has pronounced advantages over the reported functional materials; research on the scaling down of the e-synapses will serve as an interesting work in future.

We agree with the reviewer that this is a very relevant question. Therefore, we have added a brief discussion at the end of the revised manuscript to address this issue without too much deviation from the main focus of the manuscript: “The ultra-low working current of our e-

synapses is important for decreasing the energy consumption of the prospective high-density 3D synapse network. The working current of a single e-synapse partially depends on the active lateral size of the device. Therefore, scaling the lateral size down to submicrons or nanometers can decrease the working current of a single e-synapse, making an exact measurement of the working current difficult. Nevertheless, this issue can be addressed by thinning down the thickness of the active layer. In addition, the doping concentration and the type of doped ions can be engineered to maintain a reasonable working current in the range of nano-amperes.” (Paragraph 2, Page 16)

I do not clearly understand why the rectification behavior of the reported device allows to avoid cross-talk. I suppose that the "state" of the cells in the crossbar (Fig. 5-a) are read with a reading pulse in the reverse regime, but this is not clearly stated in the paper.

Response: The rectification behavior of our e-synapse is similar to that of a rectifier diode. It has been demonstrated that the crosstalk effect can be suppressed by connecting the active device to a rectifier diode (refer to *Nature Communications*, 4, 2013, 2707 and *Advanced Materials*, 24, 2012, 828–833). Clear descriptions and a modified Figure 5(a) have been presented in the revised manuscript to clarify the points. The modified top panel of Figure 5 (a) is presented below:

Typically, the parasitic paths (the **red dashed** line) for traditional crossbar memory devices without selector devices exist in parallel with the low-conductance (defined as “0” state) cell

through the neighboring high-conductance (defined as “1”) cells. However, the reverse current path (highlighted by **the dashed line circle**) can be blocked due to the rectification characteristic of the e-synapse. Thus, the device shows precise addressing (blue solid line).

We have revised the relevant content as follows: “Typically, the parasitic paths (the red dashed line in Fig. 5a) for traditional crossbar memory devices without selector devices exist in parallel with the low-conductance (defined as “0” state) cell through the neighboring high-conductance (defined as “1” state) cells. As mentioned in the previous section, the crosstalk effect can be mitigated by combining the memory device with another functional device, usually called a selector. However, there has been no mature selector available at the moment. A proper utilization of our e-synapses can directly suppress the crosstalk without the need of any selector because the reverse current path (highlighted by the dashed line circle in Fig. 5a) can be blocked due to the rectification characteristic of our e-synapse. Current-addressing test measurements were executed in order to understand this effect. Initially, one synapse was set to the “0” state and its neighboring synapses were set to “1” states. The device showed precise addressing (the blue solid line in Fig. 5a) due to the rectification characteristic of the e-synapse.” (Paragraph 2, Page 13)

But, in that case, it is not clear from data in figure 1 that a synaptic behavior is present in the reverse regime.

Response: The I - V curve of our e-synapse is asymmetric, with a remarkable rectification characteristic, as shown in Fig. 1e. Because of the remarkable rectification characteristic, the crosstalk effect can be suppressed, as discussed above. As mentioned in the above description, as the conductance level in the reverse regime (negative bias) is always negligible compared

with the forward regime (positive bias), the synaptic weight is measured as the conductance of the devices under a **positive** bias.

In conclusion, I recommend not publishing this manuscript, which is probably more relevant to a more specialist journal.

Reviewer #5 (Remarks to the Author):

In this manuscript the authors demonstrate a flexible 3D crossbar architecture. This architecture consists of selector-less memory nodes, made with a hybrid pMSSQ-based polymer. Several neuromorphic functions were demonstrated with these devices including STDP, PPF. Although the matrix material has been also used at past, from technological point of view the demonstration of a flexible 3D cross-bar architecture is still an advancement. The key feature is the selector-less memory nodes which ensure the elimination of the sneak paths from device-to-device and from layer-to-layer. I believe that the significance of the paper is more from technological point of view (i.e., realization of complex 3D architectures), than from the point of view of neuromorphic computing. In my opinion, the concept of the manuscript fits to the general scope of Nat. Comm., but there are many points/inconsistencies that have to be revised and written more carefully.

Response: we appreciate the positive comments and a great summary of the major points of our manuscript by the reviewer.

1. At the end of the abstract the authors claim that the proposed 3D geometry offers memory capability comparable to biological neural networks. At lines 50-57, p. 5 the authors use similar arguments about 3D architectures. 3D interconnectivity with cross-bar architectures can indeed offer multidimensional topologies, but it is limited to regular topologies such as multilayer nets (vector–matrix multiplication schemes in cross-bars, etc). There is a whole zoo of biological/arbitrary networks which cannot be physically realized with the cross-bar architecture. The authors could use here more realistic arguments.

Response: We appreciate the reviewer’s helpful comments and agree with the arguments. We have revised the sentence at the end of abstract as follows:

- i. “The 3D artificial synapse networks based on e-synapses enable a direct emulation of both correlated learning and trainable memory capability with strong tolerances to input faults and variations, which shows the feasibility of using them in futuristic electronic devices and can provide a new physical platform for the realization of smart memories and machine learning and for operation of the complex algorithms involving hierarchical neural networks.” (Page 2)
- ii. “In the semiconductor electronics field, the multilayer stacking architecture with a crossbar structure is an excellent candidate for realizing 3D interconnectivity.” (Paragraph 2, Page 3)

2. At lines 66-68, p. 4, the authors refer that “Even though synaptic plasticity has been extensively demonstrated, the use of one inherent characteristic of a chemical synapse, the one-direction transmission of signals, has rarely been reported”. The authors could explain

in more detail the “one-direction transmission of signals” and its implications in the neural environment.

Response: This is a very helpful suggestion to make the manuscript better understandable. We have added a brief explanation of “one-direction transmission of signals” of the chemical synapse as follows: “At a chemical synapse, one presynaptic cell releases neurotransmitter molecules into the synaptic cleft that is adjacent to another cell; then, these molecules bind to receptors on the side of the postsynaptic cell of the synaptic cleft³³. This means that the chemical synapses pass information directionally from a presynaptic cell to a postsynaptic cell, which is similar to the rectification behavior of a rectifier diode.” (Paragraph 2, Page 4)

3. At lines 92-93, p. 5, the authors refer that the dominant mechanism of the resistive switching is “still a subject of debate, the movement of the cations and their interactions are the dominant mechanisms”. The authors could be more specific here regarding the switching mechanisms.

Response: We have revised the manuscript as follows: “The nature of the resistive switching in the memristive device is still an important subject under active studies. Different models have been suggested, including the alteration of the bulk insulator’s resistivity due to the migration of ions or to a trapping-releasing process of carriers^{38,39}, the modification of the metal/insulator interface’s resistivity⁴⁰, and the formation of localized metal-atom chains bridging the electrode materials^{26,41}. Here, we attribute the conductance changes of our e-synapse to the migration of ions induced by input impulses.” (Paragraph 2, Page 6)

4. At line 113, p. 6 there is a mistake. The authors refer to “Fig. 1e” instead of “Fig. 1d”.

Response: The reviewer is correct. The *I-V* curve of a single e-synapse is shown in Fig. 1e.

5. At line 133, p. 7 the authors use the term “adaptation” for describing the modulation of the synaptic weight. This might create confusion, because adaptation is a terminology used for another biological/behavioral process.

Response: The reviewer is correct about the term “adaption” in biological process. We have corrected the sentence as follows: “The realization of variations in the synaptic weight by using an e-synapse is believed to be the most important step toward realizing other complex neurological functions through the use of neuromorphic electronics.” (Paragraph 2, Page 8)

6. At Fig. 2a, the current resolution does not seem to be enough for the measurements. Also, at Fig. 2a, the device exhibits only LTD and not LTP for positive and negative pulses. Therefore, the authors should carefully revise lines 138-143. Also, at line 143, p. 7, the authors use the term “plastic rate”. The authors should revise the term because it might lead to misunderstandings.

Response: Those are great points from the referee. We totally agree that the working current is low. However, it could be detected accurately by using our measurement equipment.

We thank the reviewer for the comments about LTD and LTP. As stated in the response to Reviewer #4, the performance of our device is different from those of most reported devices, for which positive bias leads to potentiation (or depression), and negative bias leads to depression (or potentiation). In our work, our concern is only the conductance of the devices under **positive** bias since that under negative bias is always negligible. The conductance gradually decreases with increasing number of **positive** pulses, which corresponds to the

depression process. That the conductance under **negative** bias also decreases with increasing number of **negative** pulses is true. After the application of enough **negative** pulses, a **positive** bias (instead of a negative pulse) is applied to measure the conductance state, and the device is found to have already returned to its original high-conductance state. This means that the application of **negative** biases is actually related to the potentiation process for increasing the conductance under **positive** bias, which is analogous to the **potentiation** process. To make the sentences clearer, we have rephrased the sentences as follows: “It is worth noting that the conductance of the e-synapse should be measured under a positive bias because of its remarkable rectification characteristics and negligible conductance under a negative bias. After the application of sufficient negative pulses, the e-synapse returns to its original high-conductance state measured under positive pulses. This means that the application of a negative bias is actually a potentiation process, in which the conductance increases if measured under a positive bias, which is analogous to LTP plasticity.” (Paragraph 2, Page 8)

Thank you for your useful advice. We have corrected the term “plastic rate” as “variation in the conductance” (Paragraph 2, Page 8)

7. At Fig. 3, the authors demonstrate STDP with overlapping pulses. What happens when using non-overlapping pulses?

Response: We appreciate the reviewer’s useful question. In our device, overlapping pulses should be used to realize the demonstration of the STDP behavior because no significant relaxation dynamics is observed in those devices. The utilization of overlapping pulses to achieve STFP is also described in other publications in the literature (Scientific Reports, 3, 2013, 1619; Scientific Reports, 4, 2014, 4906; IEEE Transactions on Electron Devices, 58, 2011, 2729–2737; Adv. Funct. Mater. 22, 2012, 759–2765.)

8. At Fig. 5, the authors demonstrate that the elements of each layer can be programmed independently. Are there any sneak paths across different layers? If not, is this beneficial for neuromorphic architectures? In the case of “isolated” layers, how can the information flow in 3D space be achieved?

Response: We appreciate the reviewer’s important questions. Only the selected layer was addressed in the programming process demonstrated in this work; the electrodes in the other unselected layers were floating. In this case, no sneak paths across different layers are possible. However, the output signals obtained by addressing the two or three neighboring layers are the sum of all input signals across all the addressed e-synapses distributed in the addressed 3D space. In this case, the 3D synapse networks can operate a more complex algorithm for the hierarchical neural networks, which is a meaningful and important result for future work.

We have avoided the possibility of this sentence being confusing as follows: “Note that only the selected layer was addressed and that the electrodes in other layers were floating in this programming process.” (Paragraph 1, Page 15).

Moreover, we believe potential readers will have the same question. Therefore, we have added a related discussion briefly at the end of the revised manuscript as follows: “Furthermore, by addressing two or three neighboring layers, the output signals are the sum of all the input signals across all the addressed e-synapses distributed in the addressed 3D space. In this case, our 3D-ASN provides a new physical platform for operating a complex algorithm for hierarchical neural networks.” (Paragraph 1, Page 17)

Point-by-point responses to the reviewers' comments (comments in italics, responses in blue, and content corrections in yellow)

Reviewers' comments:

Reviewer #5 (Remarks to the Author):

The authors addressed adequately most of my comments. The manuscript has now reached a satisfactory level for publication in Nat. Commun. However, a few question rise through the revision process and the authors may want to address them in order to further improve the quality of the manuscript, or even stimulate future directions. More specifically:

1. Have the authors studied the long-term lateral Cu^+ diffusion from cell to cell through stress measurements? This might be important for the long-term stability of their architecture.

Response: We thank the reviewer for sharing his/her concern. It is true that Cu ions can diffuse along both the lateral and the longitudinal directions, which could affect the long-term stability of the device. Therefore, both the effects of lateral and longitudinal diffusions actually have already been considered in previous stability measurements. As shown in the previous stability measurements (see Supplementary Fig. 5), the e-synapse exhibits long-term potentiation/depression characteristics, indicating that neither the lateral nor the longitudinal diffusion affects the long-term stability of the e-synapse. However, when the thickness of the active layer and the lateral distance between cells are decreased further to realize high density integration, the effects of lateral and longitudinal diffusions of the Cu ions on the long-term stability should be considered.

We agree with the reviewer's comments that this issue is important for further promoting device performances. Therefore, we have added a brief discussion at the end of the revised manuscript to address this issue without too much deviation from the main focus of the manuscript: "The conductance change of our e-synapse is attributed to the migration of Cu ions induced by an external electric field. Worth noting is that the Cu ions can diffuse along both the lateral and the longitudinal directions due to the difference in the doping concentration of the Cu ions, and such diffusions could affect the long-term stability of the device. As shown in the stability measurements (Supplementary Fig. 5), the e-synapse exhibits long-term potentiation/depression characteristics, indicating that neither the lateral nor the longitudinal diffusion has substantially affected the long-term stability of the e-synapse. However, when the thickness of the active layer and the lateral distance between cells are aggressively reduced further to obtain an ultra-high density integration, the effects of lateral and longitudinal diffusions of the Cu ions on the long-term stability may not be negligible." (First paragraph of Discussion, page 17)

2. A slightly better explanation is needed for the layer to layer communication. At the revision of p. 17, the authors refer that "by addressing two or three neighboring layers, the output signals are the sum of all the input signals across all the addressed e-synapses distributed in the addressed 3D space." Regarding Figs c-g, if I understand correctly, by addressing three cells from neighboring layers (for example cell (1,4) from layers 1 and 2 and 3), the output will be overall (arithmetic) sum? The authors could clarify this delicate point.

Response: This is a great question, which has helped us to further clarify an important point of our device. We believe that this question is closely related to knowledge about artificial neural network algorithms. Therefore, we have revised the sentences in question at the end of

the revised manuscript to address this issue without distracting too much from its focus:

“Furthermore, by addressing two or three neighboring layers, the measured output currents are the arithmetic sums of all the currents across the addressed e-synapses distributed in the addressed 3D space. In other words, various e-synapses can be regarded as having different weights corresponding to their conductances. Thus, the output signals are the arithmetic sums of all the weighted input signals⁴⁷.” (Paragraph 2, page 18)